# Activation of Neuroprotective Microglia and Astrocytes at the Lesion Site and in the Adjacent Segments Is Crucial for Spontaneous Locomotor Recovery after Spinal Cord Injury

**DOI:** 10.3390/cells10081943

**Published:** 2021-07-30

**Authors:** Alexandra Kisucká, Katarína Bimbová, Mária Bačová, Ján Gálik, Nadežda Lukáčová

**Affiliations:** Institute of Neurobiology of Biomedical Research Centre of Slovak Academy of Sciences, Soltesovej 4, 040 01 Kosice, Slovakia; kisucka@saske.sk (A.K.); bimbova@saske.sk (K.B.); bacova@saske.sk (M.B.); galik@saske.sk (J.G.)

**Keywords:** microglia/macrophages, microglial phenotypes: M1, M2a and M2c, A1 and A2 reactive astrocytes, Th9 compression, gene expression

## Abstract

Microglia and astrocytes play an important role in the regulation of immune responses under various pathological conditions. To detect environmental cues associated with the transformation of reactive microglia (M1) and astrocytes (A1) into their polarization states (anti-inflammatory M2 and A2 phenotypes), we studied time-dependent gene expression in naive and injured spinal cord. The relationship between astrocytes and microglia and their polarization states were studied in a rat model after Th9 compression (40 g/15 min) in acute and subacute stages at the lesion site, and both cranially and caudally. The gene expression of microglia/macrophages and M1 microglia was strongly up-regulated at the lesion site and caudally one week after SCI, and attenuated after two weeks post-SCI. GFAP and S100B, and A1 astrocytes were profoundly expressed predominantly two weeks post-SCI at lesion site and cranially. Gene expression of anti-inflammatory M2a microglia (CD206, CHICHI, IL1rn, Arg-1), M2c microglia (TGF-β, SOCS3, IL4R α) and A2 astrocytes (Tgm1, Ptx3, CD109) was greatly activated at the lesion site one week post-SCI. In addition, we observed positive correlation between neurological outcome and expression of M2a, M2c, and A2 markers. Our findings indicate that the first week post-injury is critical for modulation of reactive microglia/astrocytes into their neuroprotective phenotypes.

## 1. Introduction

Crosstalk between glial cells, microglia, and astrocytes is very important, not only for physiological conditions but also in the development and progression of diverse neuropathology [1]. Traumatic spinal cord injury (SCI) is a devastating neurological condition which results in a dysregulated microenvironment within the lesion site, and is largely driven by the immediate response of microglia and resident astrocytes [2]. In recent years several studies have reported that microglia provide the earliest response to pathological stimuli, and this microglial reaction is followed by astrocyte activation. Similarly, through their secreted molecules astrocytes regulate microglial phenotypes and functions ranging from motility to phagocytosis [1,3,4]. Both microglia and astrocytes release diverse signaling molecules to provide auto-regulatory feedback or establish their molecular microglia–astrocyte conversation [5].

Currently, considerable attention is focused on the glial scar as a major component of the SCI lesion, and on its dual role in SCI recovery. The glial scar becomes a dense limiting border around the lesion core (known as the fibrotic scar), formed predominately by scar-forming astrocytes after SCI, and it plays a very important role in spinal cord regeneration. For a long time, astrocytes were considered beneficial, because they are involved in many biological activities such as regulation of the blood–brain barrier, synaptic function and glutamate uptake [6,7,8,9,10]. However, in recent decades, the crucial role of astrocytes in pathological processes has been revealed, including neurodegenerative diseases, stroke and CNS injuries [11,12,13,14,15]. After SCI, astrocytes are activated and some of them rapidly proliferate to form an astrocytic scar border. Glial scar formation is a highly dynamic process with multiple cellular phenotypes (such as astrocytes, microglia and oligodendrocyte precursor cells), which not only directly contribute to the dual role of the glial scar, but also interact with other components, increasing its complexity. It is very difficult therefore to target the glial scar for therapeutic purposes [16]. Knowing the precise timeline of glial scar formation in each experimental model is crucial for making the right decision to target the treatment at a suitable moment after SCI, because radical interventions in glial scar formation too soon after SCI may not contribute to recovery [17,18].

As mentioned earlier, macrophages, activated microglia and infiltrated monocytes are major players in neuroinflammation [19]. We and others have shown that acute depletion of macrophages is neuroprotective and promotes functional recovery after SCI [20,21]. In the present study, compared with other SCI research we focus on some novel aspects, which have not been examined so far. In a model of Th9 compression (40 g/15 min) in rats, we investigated the gene expression for microglia/macrophages (Iba1, CD11b, Cx3Cr1), reactive M1 microglia (IL-1β, CD68, iNOS, IL-6), common astrocytes (GFAP, S100B, Lcn2, Serpina3n) and reactive A1 astrocytes (C3, TNF-α, C1q) in the injured spinal cord at one and two weeks after SCI, i.e., at time points which are key for activation of pro-inflammatory vs. anti-inflammatory macrophage and astrocyte phenotypes. In addition, we examined (i) the gene expression of neuroprotective markers M2a (CD206, Ym1, IL1rn, Arg-1), M2c (TGF-β, SOCS3, IL4R α) and A2 (Tgm1, Ptx3, CD109) which could limit secondary inflammatory-mediated injury and/or promote CNS repair; (ii) the ratios M1/M2 and A1/A2, which could have significant implications for SCI; and (iii) the correlation between the neurological outcome after SCI and the expression of genes associated with neuroprotective phenotypes (M2a, M2c and A2). These genes express anti-inflammatory and immune-regulatory molecules important for repair, regeneration and wound healing in time-dependent manner. All these changes were detected at the lesion site and 3 mm craniocaudally from the injury epicenter. We found positive correlation between the neurological score and the expression of M2c marker, TGF-β at the lesion site at one week (R^2^ = 0.872), followed by mild positive correlation both at the lesion site and below it at two weeks post-SCI. Similar but moderate correlation was noted between the neurological outcome and the expression of SOCS3 (M2c marker), and CD109 and Ptx3 (A2 markers). The expression of Tgm1 and Ptx3 (A2 markers) mildly correlated with neurological outcome at the lesion site and in the segment above it, while CD109 showed mild positive correlation at the lesion site and in the adjacent caudal segment. We demonstrate that activation of these neuroprotective markers between the first and second week post-SCI is required for spontaneous locomotor recovery.

## 2. Materials and Methods

### 2.1. Experimental Animals

A total of 15 adult female Wistar rats weighing 250 to 300 g were used for the experiment. The rats were housed individually in clear plastic cages on a 12-h dark/light cycle in a temperature- and humidity-controlled environment. Food and water were available ad libitum. The experiments were approved by the State Veterinary and Food Administration in Bratislava (decision No. 4434/16-221/3), as well as by the Animal Use Committee at the Institute of Neurobiology, Biomedical Research Center of Slovak Academy of Sciences, in accordance with the EC Council Directive (2010/63/EU) regarding the use of animals in scientific research. The animals were divided into three experimental groups as follows: (1) control animals (*n* = 5); (2) animals subjected to Th9 compression surviving for 7 days (*n* = 5) and (3) those surviving for 14 days (*n* = 5)**.**

### 2.2. Spinal Cord Compression

Spinal cord compression was performed under isoflurane anesthesia (2–4%; AbbVie, Bratislava, Slovak Republic; in 1.5–2.0 L/min oxygen), delivered by mask. The back of the rats was shaved and cleaned with Betadine (10%; Egis Pharmaceuticals Plc, Budapest, Hungary). The spinal cord was exposed very carefully at low thoracic level (Th9) and compressed using a compression device with a weight of 40 g for 15 min. The body temperature was maintained at 37 °C during the whole surgical procedure. Postoperative care began immediately after surgery with Amoksiklav antibiotic (Sandoz Pharmaceuticals, Ljubljana, Slovenia; 30 mg/kg; i.m.) and Novasul analgesic (Richterpharma; Wels, Austria; 2 mL/kg, i.m.) for three days. After SCI the rats’ bladders were expressed twice a day until their bladder reflexes were restored.

### 2.3. Analysis of Gene Expression Using RT-PCR

At the end of survival, the experimental animals were deeply anesthetized with thiopental (50 mg/kg, i.p.) and the backbones were exposed and cut at Th7–Th10 segmental level. The epicenter of spinal cord injury (0.3 cm section) and spinal cord blocks 0.3 cm cranially (termed as +1) and caudally (termed as −1) from the epicenter of the injury were rapidly dissected, carefully frozen with liquid nitrogen and stored at −70°C until processing.

Tissue was thawed on ice and total RNA was extracted from spinal cord segments using Genezol Reagent (Geneaid, New Taipei City, Taiwan) according to the manufacturer’s manual (Geneaid). RNA quantity and quality were checked using spectrophotometric analysis, measuring the 260/280 ratio (NanoDrop2000 c, Thermo Fisher Scientific, Waltham, MA, USA). Reverse transcription of 2000 ng of total RNA was performed in a final volume of 20 μL using a High-Capacity cDNA Reverse Transcription Kit (AB applied biosystems by ThermoFisher Scientific, Waltham, MA, USA) and a T1000^TM^ Thermal Cycler (Bio-Rad, Hercules, CA, USA) during one cycle: at 25 °C for 10 min, at 37 °C for 2 h, and at 85 °C for 5 min with subsequent cooling to 4 °C. cDNA samples were stored at −20 °C until processing.

The amplification of cDNA (10 ng per reaction) was performed using the CFX96^TM^ Real-Time System (Bio-Rad, Hercules, CA, USA). The primers (see Table 1) were designed using Geneious (Biomatters, Ltd., Auckland, New Zealand) software. For RT-PCR, PowerUp^TM^ SYBR^TM^ Green Master Mix (AB applied biosystems by ThermoFisher Scientific, Waltham, MA, USA) was used with 1 µM concentration of each primer. The RT-PCR reaction protocol was as follows: 10 min at 95 °C followed by 50 cycles consisting of 15 s of denaturation at 95 °C and annealing/extension for 1 min at 60 °C. The expression of the analyzed genes was normalized to the housekeeping gene 18S. The relative level of mRNA was calculated using the ΔΔC_t_ method. All analyses were duplicated.

### 2.4. Behavioral Model (BBB Scale)

The rats’ post-injury behavior was assessed using the Basso, Beattie and Bresnahan (BBB) locomotor scale method. Functional recovery was monitored by evaluating the toe spread, righting reflex, withdrawal reflex (to extension) and contact placing test. The rating scale represents sequential recovery stages from complete paralysis (zero) to normal movement (21 points) [22] and categorizes combinations of rat joint movement, hindlimb movements, stepping, forelimb and hindlimb coordination, trunk position and stability, paw placement and tail position. All functional scores were obtained on days 1, 2, 3, 5, 7, 10 and 14.

### 2.5. Statistical Analysis

The data from the RT-PCR were analyzed with an unpaired *t*-test using GraphPad Prism version 6.01 (San Diego, CA, USA) and were expressed as mean values with standard deviation (SD). The significance level was set as *p*-value less than 0.05. Correlation analyses were performed with the Pearson correlation test.

## 3. Results

### 3.1. Infammatory Response after SCI

In in vivo conditions, it is very difficult to distinguish microglia from monocyte-derived macrophages (MDMs) which enter the CNS from the peripheral blood after injury and adopt many of the markers and behaviors of microglia, as well as from macrophages which populate neuronal space in healthy or pathological conditions [23]. Taking this fact into consideration, we first investigated changes in the expression of genes associated with microglia/macrophages (Iba1, CD11b, Cx3Cr1) in the naive spinal cord parenchyma, at the lesion site and in surrounding spinal cord tissue one and two weeks after Th9 compression. As expected, the mRNA expression of all these markers was substantially overexpressed at the lesion site and in spinal cord tissue immediately adjacent to the lesion one week post-SCI, compared to naive controls (Figure 1a). Interestingly, the expression of CD11b, a cell surface marker of microglial cells, was more than three-fold higher than the expression of Iba1 or Cx3Cr1. The response of this marker to SCI also differed in the later post-SCI period, when CD11b mRNA decreased significantly through the whole studied area (*p* ˂ 0.0001), while Cx3Cr1 was down-regulated (*p* ˂ 0.001) only in the segment below the lesion site. On the other hand, Iba1, a commonly used marker of microglia, was strongly elevated (*p* ˂ 0.001) two weeks post-SCI in the cranial segment. No significant alterations were seen at the site of injury and caudally, where Iba1 expression remained at the same level during both survival times.

Pro-inflammatory molecules produced predominantly by classically activated M1 microglia play a key role in the activation of downstream pathways, and lead to tissue damage [24]. To study the activation of M1 microglia in the subacute phase of SCI, we next studied changes in the expression of pro-inflammatory cytokines (IL-1β, IL-6), an activated macrophage marker (CD68) and iNOS, an oxidative stress marker (Figure 1b). One week after the SCI, the level of CD68 mRNA expression was substantially higher than in naive controls, showing ~5–8-fold increase through the whole rostrocaudal extent. Similarly, IL-6 mRNA expression increased 4-fold at the lesion site and 2.7-fold caudally, whereas IL-1β and iNOS were overexpressed only at the epicenter of SCI (1.9- and 3-fold). Compared to controls, both pro-inflammatory cytokines (IL-1β, IL-6) and iNOS remained without changes one week post-SCI in the cranial segment, which is known to host a neuroprotective and neurotrophic environment. The results show that M1 markers were expressed during the first post-SCI period in a regionally dependent manner. One week later, all pro-inflammatory markers significantly increased their expression in the proximal segment (IL1β—*p* ˂ 0.0001; IL-6—*p* ˂ 0.01; CD68—*p* ˂ 0.05 and iNOS—*p* ˂ 0.001) compared to the early post-SCI point. All these data indicate that activated macrophages (CD68 marker) are recruited at and around the lesion site, where they accumulate extensively at both post-SCI time points and adopt an activated state which could be eventually partially resolved during the intermediate and/or chronic phases. IL-6 and iNOS, however, significantly decreased their expression two weeks post-SCI at the lesion site and caudally compared to the early time period, suggesting a partial return to homeostasis and/or their participation in the induction of reactive A1 phenotypes [3] or M1/M2 polarization.

Astrocytes surrounding the epicenter of injury become reactive; they migrate to the lesion epicenter and play an important role in the SCI environment [25]. To distinguish the individual subtypes of reactive astrocytes, we analyzed seven genes which modulate astrogliosis in the SCI environment (GFAP and S100B—classical reactive markers; C3, TNF-α and C1q—markers of neurotoxic A1 astrocytes; Lcn2 and Serpina3n—markers of acute and persistent reactive astrogliosis) (Figure 2 and Figure 3).

The results show that SCI induced time-dependent astrogliosis which also varied with distance from the epicenter of the lesion. One week post-SCI, the expression of common astroglial markers (GFAP and S100B) was reduced predominantly in the lesion core, suggesting that this decrease could be consistent with rapid astrocyte death (Figure 2a) [26]. The expression of both these markers increased more than three-fold at the lesion site and in cranial segments two weeks post-SCI. Liddelow et al. [3] reported that A1 phenotypes, induced by releasing cytokines, lose their normal astrocyte functions and gain new neurotoxic functions. We confirmed that A1 markers (C3, TNF-α and C1q) were up-regulated within the lesion site and in the surrounding area one week post-SCI relative to naive controls. A significant increase in TNF-α mRNA and C1q mRNA was also observed in the cranial segment at two weeks post-SCI relative to the early post-SCI period (Figure 2b). The expression of TNF-α, which was strongly produced by macrophages/monocytes during acute inflammation in the epicenter of injury, significantly decreased (*p* ˂ 0.05) two weeks post-injury. A similar trend was noted in the expression of C1q in the caudal segment. This marker is responsible for a diverse range of signaling events within cells leading to necrosis or apoptosis. Although the proliferation and transformation processes of reactive astrogliosis were very intensive within both time periods, our results show that the expression of astrocytes was dramatically increased predominantly cranially and at the lesion site two weeks after injury. A secreted lipophilic protein (Lcn2) was strongly expressed two weeks after spinal cord insult at the epicenter of injury (4.7-fold) and in the cranial segment (4.2-fold) (Figure 3), whereas Serpina3n, a secreted peptidase inhibitor whose expression is induced by inflammation [27], was overexpressed (1.5-fold) in the same time period only at the epicenter of injury.

It is not clear from these results which polarization state of microglia and astrocytes was impacted after SCI. Given the fact that upon phenotypic polarization both microglia and astrocytes could have neurotoxic or neuroprotective function, we next carried out a more detailed analysis of anti-inflammatory M2a and M2c microglia, and A2 astrocytes.

### 3.2. Neuroprotective Environment after SCI

M2 microglia show neuroprotective effects through tissue repair and toxicity clearance in many CNS diseases [28,29]. In the last two decades, activated M2 microglia have been gradually categorized into four subtypes (M2a, M2b, M2c and M2d). All these phenotypes are considered to be anti-inflammatory repair M2 microglial cells, but there are several differences which could be characterized by changes in expression of the relevant markers [30,31]. In the present study, we turned our attention to M2a microglia (CD206, Ym1, IL1rn, and Arg-1 markers) playing a key role predominantly in cell repair and regeneration by expressing anti-inflammatory and immune-regulatory molecules, and to the M2c microglial phenotype (TGF-β, SOCS3 and IL4R α markers), which is largely phagocytic and plays an important role in wound healing. Increased expression of mRNA markers typical for the M2a phenotype (CD206 and IL1rn) and all M2c markers was found in the whole cranio–caudal extent of the spinal cord one week after SCI (Figure 4a,b).

In the later post-SCI period, the levels of all M2a and M2c markers significantly increased in the cranial segment compared to the one week survival interval. This insight into regional heterogeneity of neuroprotective microglial phenotypes shows that the expression of CD206, IL1rn and Arg-1 (M2a phenotype), and SOCS3 and IL4R α (M2c phenotype) was significantly down-regulated at the lesion site and/or in the caudal segment two weeks post-SCI. These data correlate with the detrimental microenvironment seen at the lesion site and in the spinal cord parenchyma surrounding the injury. We also found that neuroprotective microglia proliferated extensively during the first two weeks post-SCI, depending on provided function.

The neuroprotective A2 astrocyte phenotype up-regulates many neurotrophic factors after SCI. The mRNA expression level of two A2 markers (Tgm1 and CD109) was strongly activated at the lesion site at both time points (Figure 5). A clear time-dependent difference in gene expression of all A2 markers (Tgm1, CD109, Ptx3) was further demonstrated in the cranial segment. These results show that the expression of these genes was significantly increased above the site of injury two weeks post-SCI. In addition, Ptx3, which is known to be rapidly produced and released by several cell types (in particular by mononuclear phagocytes, fibroblasts and endothelial cells in response to primary inflammatory signals), was also overexpressed at the lesion site two weeks post-SCI vs. the early post-SCI period (*p* ˂ 0.01). A significant decrease in the mRNA expression of Tgm1 was identified only in the caudal segment in the later post-SCI period (*p* ˂ 0.05). The heat map of genes at our focus of interest presents a comprehensive picture of the post-SCI microenvironment at the lesion site, cranially and caudally at both time points (Figure 6).

### 3.3. Microglia and Astrocyte Phenotypic Polarization and Its Impact on Neurotoxic or Neuroprotetive Functions

The interactions between reactive microglia and astrocytes in the environment of the injured CNS are still unexplored. To find out whether activation of microglia by pro-inflammatory mediators can convert astrocytes to the neurotoxic A1 phenotype within two weeks after SCI, we compared the ratio between overall gene expression of M1 and A1 markers (Figure 7a) and the strongest representative markers (CD68 and TNF-α) (Figure 7b) at the lesion site and in the surrounding areas. Although the overall M1 expression significantly exceeded the expression of A1 (*p* ˂ 0.0001) in the segment located above the site of injury two weeks post-SCI, the prevalence of CD68 (M1) above the A1 marker (TNF-α) was found in the same area at the earlier post-SCI point (Figure 7a,b). The ratio between CD68 and TNF-α (M1/A1) also revealed that microglia were switched to the M1 phenotype predominantly at the lesion site and below the epicenter of injury two weeks post-SCI. We suggest that the highly detrimental microenvironment in these regions does not create a suitable milieu for modulation of M1 microglia to A1 astrocytes in the later post-SCI period.

Next, we examined the ratios between the pro- and anti-inflammatory markers within one phenotype (M1/M2 microglia) and (A1/A2 astroglia) (Figure 7c,e), and their strongest representative markers (CD68/CD206) and (TNF-α and CD101) (Figure 7d,f). The ratios of overall M1/M2 or A1/A2 polarization did not reveal any significant differences between the spinal cord regions and/or time points. Two weeks post-SCI, the CD68/CD206 (M1/M2) ratio was in favor of neurotoxic microglial markers at the lesion site. TNF-α, a neurotoxic astroglial marker, prevailed over CD101 (A2 marker) in the same spinal cord region one week post-SCI.

### 3.4. Correlation between Neurological Score and Expression of Neuroprotective Microglia and Astroglia Phenotypes

The recovery stages for rat hindlimb motor function were monitored for two weeks post-SCI using the Basso–Beattie–Bresnahan (BBB) locomotor score scale. One day post-injury, all the animals suffered from complete paraplegia (0.3 ± 0.42). Their motor function started improving soon after SCI, but more progressive spontaneous improvement of locomotor function was seen from day 7 to day 14 (2.750 ± 0.08 and 7.5 ± 0.35) (Figure 8).

Pearson´s correlation analysis was performed to determine the relationship between the neurological outcome and relative gene expression of markers characteristic for protective microglial M2c (Figure 9) and M2a phenotypes (Figure 10).

We found positive correlation between the neurological score and the expression of TGF-b (M2c marker) at the lesion site at one week (R^2^ = 0.872; P = 0.020), followed by mild correlation both at the lesion site (R^2^ = 0.124, P = 0.0561) and below it (R^2^ = 0.362; P = 0.283) at two weeks post-SCI (Figure 9a). The expression of SOCS3 (M2c marker) mildly correlated with the neurological outcome at the lesion site one week post-SCI, and cranially and caudally at both time points. In addition, the measurements of gene expression for IL4R α (M2c) and CD206 (M2a) markers mildly correlated with the neurological score (R^2^ = 0.350, P = 0.293; R^2^ = 0.309, P = 0.330) at the lesion site one week post-SCI (Figure 9c and Figure 10a). All these data suggest that activation of neuroprotective microglial markers between the first and second week post-SCI (lesion site, and caudally as well as cranially from the injury epicenter) is required for spontaneous locomotor recovery. Moderately positive correlation was also found between the gene expression of A2 markers (Tgm1, Ptx3 and CD109) and neurological outcomes at both time points after SCI (Figure 11).

Depending on the role of these neuroprotective astrocyte markers in the injured spinal cord parenchyma, they were positively correlated at the lesion site and cranially (Tgm1 and Ptx3) and/or at the lesion site and caudally (CD109). The correlation coefficients were higher at one week (ranging from 0.381, P = 0.268 to 0.546, P = 0.154) than two weeks post-SCI (ranging from 0.136, P = 0.542 to 0.220, P = 0.426).

## 4. Discussion

Microglia and macrophages have highly plastic phenotypes which can change rapidly in response to a variety of conditions [32]. After SCI, many factors with antagonistic pro- and anti-inflammatory properties act simultaneously on macrophages and microglia. The present study indicates that representative genes for activated macrophages/microglia and reactive M1 microglia were more highly expressed through the studied areas than reactive astrocytes (common markers, A1, Lcn2 and Serpina3n) at acute post-injury survival time (one week after SCI), and progressively decreased thereafter (two weeks after SCI) at the lesion site and below it. The increased expression of these microglia markers persisted or was significantly increased only in the adjacent cranial segment two weeks post-SCI. We also found that CD11b (macrophages/microglia), CD68 (M1) and CD206 (M2a) transcript levels were extensively increased (15.5-fold, 8.4- and 7.2-fold) predominantly at the lesion site one week post-injury, i.e., at the time point when microglia extensively proliferate and accumulate, forming a dense scar at the interface between the fibrotic scar and the yet-to-be-formed astrocytic scar, referred to as the microglial scar [33]. Recently published data indicate that microglia induce astrocyte activation and could determine the fate of astrocytes [5]. Our results are consistent with this finding, since the gene expression of almost all markers for astrocytes (common markers, A1, A2, Lcn2 and Serpina3n) was significantly increased cranially and/or at the lesion site two weeks after Th9 compression compared to one week post-SCI. It has also been reported that astrocytes have the potency to trigger microglial activation and control their cellular functions [5]. Multiple experimental studies have confirmed that the neuroinflammatory response after SCI is mediated by various cell types, specifically astrocytes, resident microglia, infiltrating immune cells, and endothelial cells which form the linings of the blood vessels [21,34,35]. Although we did not study the release of diverse signaling molecules in individual cell phenotypes, our results show that the activation of neuroprotective processes takes place at least at the lesion site and in the adjacent cranial segment one week post-injury. Neuroprotective microglia markers (M2a and M2c) were expressed at the lesion site and in adjacent areas one week post-injury, but their expression was temporal in the most vulnerable lesion site and caudal area. Interestingly, all neuroprotective microglia (M2a and M2c) and neuroprotective astroglia (A2) markers were significantly up-regulated above the lesion site at later post-SCI period. We suggest that molecular conversation among these phenotypes may occur in the cranial segment, known to host a neuroprotective environment two weeks post-SCI.

After spinal cord insult, microglia/macrophages polarize into several states: a classically activated pro-inflammatory (M1) phenotype, an alternatively activated anti-inflammatory (M2a and M2b) phenotype and M2c (acquired deactivation) phenotype [36,37]. The neurotoxic M1 sub-group of activated microglia/macrophages is activated soon after SCI, and they express high levels of pro-inflammatory cytokines [38], including IL-1β, Il6 Il-12, the well-studied TNFα (which is attributed to both neurotoxic M1 and A1 sub-groups of reactive astrocytes), oxidative metabolites, chemokines and proteases [39]. Conversely, the M2 phenotype releases anti-inflammatory cytokines and down-regulates inflammation and facilitates wound healing. Data reported earlier [19,40] indicated that macrophages/microglia in the injured spinal cord were predominantly polarized to the M1 phenotype, and after their quick activation during the first few days they remained activated for 28 days after injury. Activation of a small number of M2 macrophages/microglia was observed, but this phenotype was short-lived, dissipating within 3–7 days after injury. Our data show regionally dependent changes in expression of both pro-inflammatory M1 markers and neuroprotective M2 phenotypes one and two weeks after spinal cord injury. M1 and M2 phenotypes coexisted in the injured area within the first two weeks after SCI, with M1 prevailing significantly. M2 phenotypes persisted or decreased at the lesion site or caudally two weeks post-SCI even though after seven days the integrity of the blood–brain barrier starts to be repaired and edema begins to be resolved [41].

In the past, microglial activation in the injured CNS was perceived mainly as neurotoxic and harmful. Polymorphonuclear leukocytes (PMNLs) and monocytes in the traumatized spinal cord tissue depend upon certain endothelial adhesion molecules, such as intercellular adhesion molecule-1 (ICAM-1). This molecule influences the extravasation of leukocytes by binding to counterreceptors (CD11b) on the surface of the activated leucocytes, which results in firm adhesion between the leucocyte and the endothelium [42]. Isaksson et al. [43] reported that this adhesion is transient and is followed by migration through the endothelial barrier. Figley et al. [41] studied endogenous angiogenic response after SCI in a clip-compression model for two weeks post-injury. The authors found maximal blood–spinal cord barrier (BSCB) disintegration at the lesion site and approximately 2 mm craniocaudally from the injury epicenter one day after SCI, with significant disruption observed until five days post-SCI. We observed massive (11–15-fold) CD11 expression throughout the whole injured area (9 mm) one week after SCI. This change is consistent with microglial activation, i.e., transformation of phagocytic microglial cells, seen earlier in the white matter one day to one week after extradural compression injury at low thoracic level [43]. Gris et al. [20] and Saville et al. [44] demonstrated that anti-CD11d monoclonal antibody applied post-SCI could effectively block the entry of neutrophils and improve the neurological outcome. The expression of CD11b, which occurs constitutively on microglial cells and is up-regulated in activated microglia, decreased continuously throughout the spinal cord during the later post-SCI period. In contrast to this, the expression of Cx3Cr1 was down-regulated at this time point only below the site injury. Zhang et al. [26] reported that CX3CL1 is expressed in spinal neurons and induces microglial activation via its microglial receptor Cx3Cr1 (neuron-to-microglia signaling). Since TGF-b and SOCS3, both anti-inflammatory microglial M2c phenotypes positively correlated with functional outcome in caudal segment two weeks post-injury, we suggest that the decrease in Cx3Cr1 could be part of an intrinsic cellular response occurring during neuroregeneration. Similarly, phagocytic macrophages of microglial and monocytic origin, which are abundant at the lesion site, produce lysosomal protein CD68 [45]. We found that the expression of this marker prevailed 8-fold at the lesion epicenter and 5–6-fold in both adjacent areas across both time points. These results indicate that macrophages were still phagocytic and did not stop producing this lysosomal protein until two weeks post-injury.

Chamankhah et al. [46] studied gene expression in the spinal cord and reported the highest IL-1β transcript level at the lesion epicenter on day 3 after spinal cord clip injury. These authors confirmed the continuing up-regulation of this pro-inflammatory cytokine until eight weeks post-injury. In the present study, IL-1β mRNA was overexpressed (~1.9-fold) at the most vulnerable lesion site at both post-SCI survival times. Moreover, significant elevation of IL-1β mRNA was found in the spinal cord parenchyma above the lesion site two weeks post-SCI. As reported previously, this cytokine can directly affect glial cells, endothelial cells, and even neurons, since they all express interleukin 1 receptor type 1 (IL-1R1) [47]. The infusion of rmIL-1ra, the IL-1 receptor antagonist, for 72 h after SCI completely abolished the increases in contusion-induced apoptosis and caspase-3 activity, indicating that early IL-1β expression is detrimental. We previously reported 12-fold increase in the IL-1β level in blood serum 4 h after Th9 compression, and its rapid decrease at one day post-SCI [21]. On the other hand, Rust and Kaiser [48] reported that astrocyte activation by IL-1β can exert neuroprotective effects by stimulating the repair of the blood–brain barrier and decreasing its permeability. In our study, this M1 microglia marker was expressed at the lesion site and/or in the adjacent cranial segment, i.e., in the segments with highest expression of A2 neuroprotective astrocyte markers at both time points. Bellver-Landete et al. [33] also confirmed that proliferating activated neuroprotective microglia persist in the injured spinal cord for five weeks, and play an important role both in the scar formation which develops after SCI and in the functional recovery by protecting non-injured neurons and oligodendrocytes from inflammation-mediated tissue damage.

Scar-forming astrocytes are another very important component of the post-SCI microenvironment with a dual role in the pathological process of SCI, both protective and inhibitory. Reactive astrocytes interact extensively with reactive microglia during glial scar formation [4]. After activation, microglia release cytokines which trigger and maintain the activation of astrocytes. We show that cranially to SCI, astrocyte activation (GFAP, S100B, TNF-α and C1q) was concomitant with activation of Iba-1 in the later post-SCI period. Bellver-Landete et al. [33] showed that the release of typical insulin-like growth factor 1 (IGF-1) by microglia induces glial scar formation and triggers the activation and proliferation of astrocytes, while the in vivo inhabitation of IGF-1 reduces astrocytic proliferation and migration to the lesion, just as the depletion of microglia also leads to decreased glial scar formation and worse functional recovery. The interactions between reactive microglia and reactive astrocytes have also been demonstrated by the finding that microglia may transform the neuroprotective astrocyte phenotype via down-regulation of the P2Y_1_ purinergic receptor [4]. Only a few reports indicate that microglia are responsible for inducing a neurotoxic astrocyte phenotype [3]. Our study reveals the behavior of microglia and astrocytes in the most important time intervals after SCI. Astrocytes are the principal type of cells undergoing necrosis after SCI, but they also persist in the injured spinal cord for two weeks [49]. This is partially confirmed by our data, showing that markers of reactive astrocytes in contrast to reactive microglia are expressed at the site of injury and cranially two weeks after Th9 compression. This statement was confirmed not only by GFAP and S100B, which may not be the most accurate markers of astrocyte activity, but also by Lcn2 and Serpina3n, markers which have some more specific hallmarks of reactive astrocytes. Ren et al. [50] indicated that from two to several weeks after SCI, the scar-forming astrocytes finish their phenotypic change, and the glial scar becomes completely mature. Thereafter, the spinal cord lesion is stabilized and a long chronic phase of regeneration begins. We observed that the neurotoxic A1 phenotype predominated at the lesion site one week after SCI.

Since inflammation and astrogliosis play critical roles in cavity formation after SCI, the beneficial effects of neurotoxic environment inhibition on SCI may come from the microglia/macrophages and astrocytes reactive polarization states. This is inevitable because the glial scar plays a dual role. Recently, Yang et al. [16] reported that the integrity of the glial scar provides a critical barrier limiting the spread of inflammation; however, they also observed detrimental effects of fibrotic tissue and macrophages in the chronic phase. Inhibiting M1/A1 and/or promoting M2 and A2 polarization could be a very effective treatment strategy to improve functional recovery after SCI. However, manipulation of the glial scar should be moderate, avoiding compromising its integrity and adversely affecting its post-injury support functions. We suggest that precise identification markers affecting microglia–astrocyte conversion in the detrimental environment after SCI is of key importance for future therapeutic studies.

## Figures and Tables

**Figure 1 cells-10-01943-f001:**
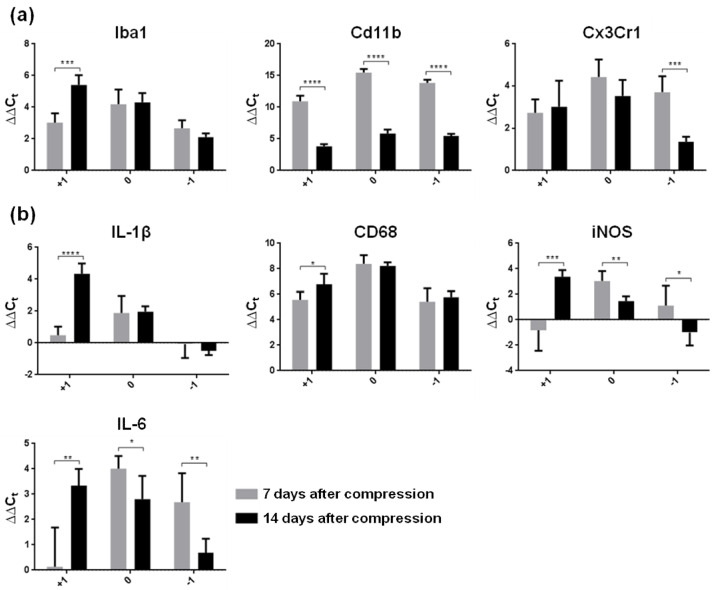
Gene expression of reactive microglial markers in spinal cord segments (0—lesion site, +1 cranially, −1 caudally) one and two weeks after Th9 compression. (**a**) Microglia/macrophage markers (Iba1, Cd11b, Cx3Cr1), (**b**) markers representing M1 phenotype of reactive microglia showing inflammation (IL-1β, CD68, IL-6) and oxidative stress (iNOS). Data are mean values of five experiments ±SD. The mRNA levels for each marker were normalized to the mRNA levels of 18sRNA and expressed as ∆∆C_t_ values relative to controls. Results were statistically evaluated using unpaired *t*-test; * *p* ˂ 0.05; ** *p* ˂ 0.01; *** *p* ˂ 0.001; **** *p* ˂ 0.0001. Iba1—ionized calcium-binding adaptor molecule 1; Cd11b—beta-integrin marker of microglia; Cx3Cr1—fractalkine receptor; IL-1β—interleukin-1β; CD68—cluster of differentiation 68; iNOS—inducible nitric oxide synthase, IL-6—interleukin-6.

**Figure 2 cells-10-01943-f002:**
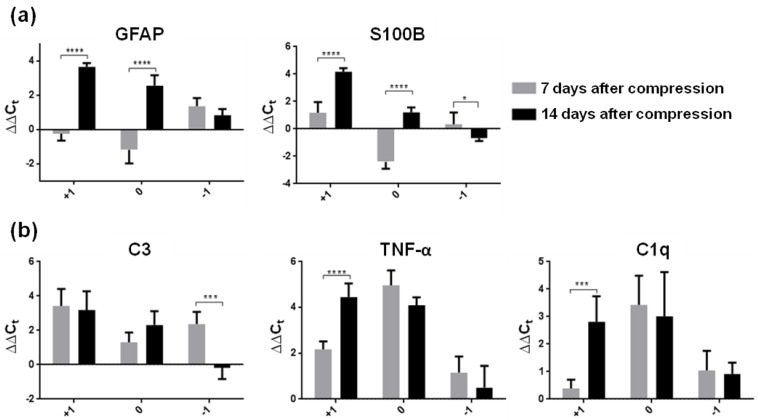
(**a**) Relative expression of common reactive astrocyte genes (GFAP and S100B) in the spinal cord segments (site of the injury, cranially and caudally) one and two weeks after Th9 compression. Gene expression of both markers was strongly up-regulated mainly at the injury site and adjacent cranial segments two weeks after SCI. (**b**) Expression of genes (C3, TNF-α and C1q) induced by neurotoxic A1 reactive astrocytes one and two weeks after Th9 compression. Data are mean values of five experiments ±SD. The mRNA levels for each marker were normalized to the mRNA levels of 18sRNA and expressed as ∆∆C_t_ values relative to controls. Results were statistically evaluated using unpaired *t*-test; * *p* ˂ 0.05; *** *p* ˂ 0.001; **** *p* ˂ 0.0001. GFAP—Glial fibrillary acidic protein; S100B—Calcium-binding protein B; C—Complement 3; SCI—spinal cord injury; TNF-α—tumor necrosis factor-α; C1q—Complement component 1q.

**Figure 3 cells-10-01943-f003:**
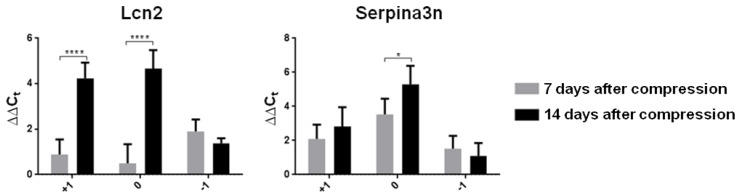
Expression of universal markers (Lcn2 and Serpina3n) for reactive astrocytes in spinal cord segments one and two weeks after Th9 compression. Lcn2 represents the earliest part of the reactive astrogliosis response, and conversely, Serpina3n is a useful marker of more persistent reactive gliosis response. Data are mean values of five experiments ±SD. The mRNA levels for each marker were normalized to the mRNA levels of 18sRNA and expressed as ∆∆C_t_ values relative to controls. Results were statistically evaluated using unpaired *t*-test; * *p* ˂ 0.05; **** *p* ˂ 0.0001. Lcn2—Lipocalin 2; Serpina3n—Serine (or cysteine) peptidase inhibitor.

**Figure 4 cells-10-01943-f004:**
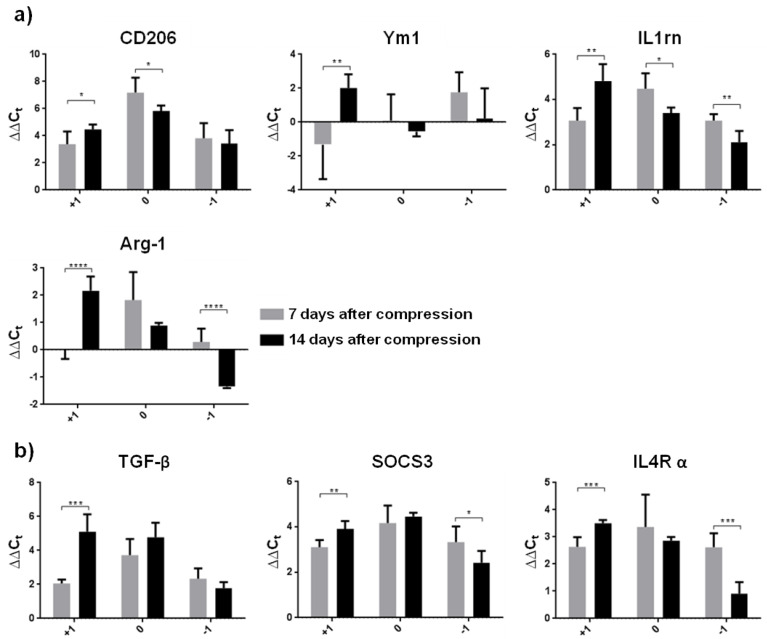
Graphs demonstrating relative gene expression of neuroprotective microglia in the spinal cord (injury site, cranially and caudally) one and two weeks after Th9 compression. (**a**) M2a subtype of activated microglia represented by CD206, Ym1, IL1rn and Arg-1; (**b**) M2c subtype of activated microglia markers (TGF-β, SOCS3 and IL4R α). Data are mean values of five experiments ±SD. The mRNA levels for each marker were normalized to the mRNA levels of 18sRNA and expressed as ∆∆C_t_ values relative to controls. Results were statistically evaluated using unpaired *t*-test; * *p* ˂ 0.05; ** *p* ˂ 0.01; *** *p* ˂ 0.001; **** *p* ˂ 0.0001. CD206—mannose receptor and C-type lectin; Ym1—chitinase-like protein-1; IL1rn—interleukin 1 receptor antagonist; Arg-1—arginase-1; TGF-β—transforming growth factor beta; SOCS3—suppressor of cytokine signaling 3; IL4R α—interleukin 4 receptor alpha.

**Figure 5 cells-10-01943-f005:**
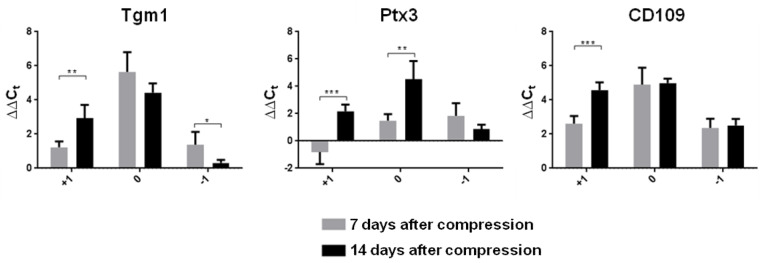
Expression of genes induced by neuroprotective A2 astrocytes one and two weeks after Th9 compression. Data are mean values of five experiments ±SD. The mRNA levels for each marker were normalized to the mRNA levels of 18sRNA and expressed as ∆∆C_t_ values relative to controls. Results were statistically evaluated using unpaired *t*-test; * *p* ˂ 0.05; ** *p* ˂ 0.01; *** *p* ˂ 0.001. Tgm1—Transglutaminase 1; Ptx3—Pentraxin 3; CD109—Cluster of differentiation 109.

**Figure 6 cells-10-01943-f006:**
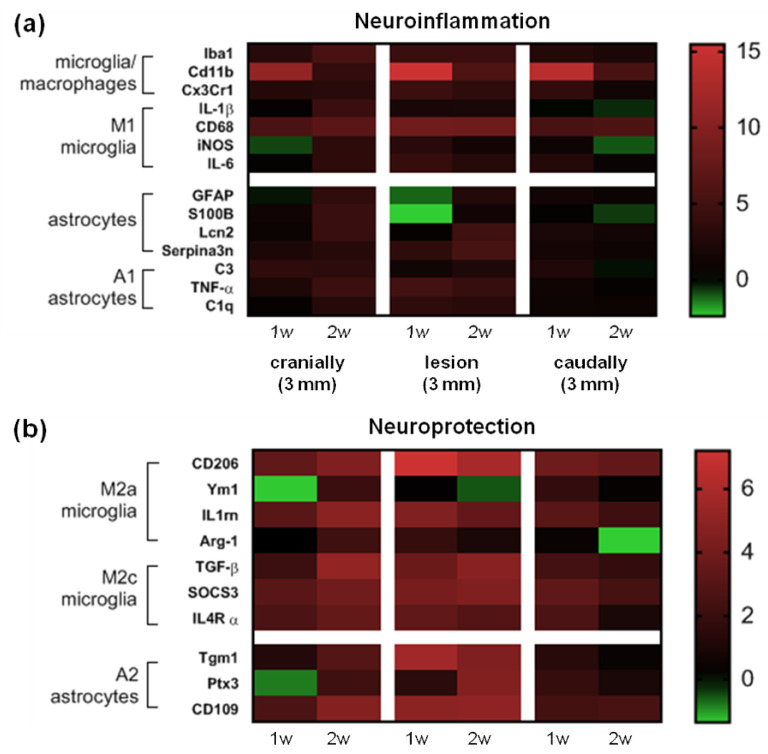
Heat map of genes belonging to the neuroinflammatory (**a**) and neuroprotective (**b**) microglia/macrophages and astrocytes. The horizontal axis represents time following Th9 compression at the lesion site, then cranially and caudally and the vertical axis represents specific genes. Black indicates no change in relative ∆∆C_t_ values relative to controls, whereas shades of green or red represent a decrease or increase of marker values. Imaging software: GraphPad Prism (version 8; GraphPad Software Inc., San Diego, CA, USA).

**Figure 7 cells-10-01943-f007:**
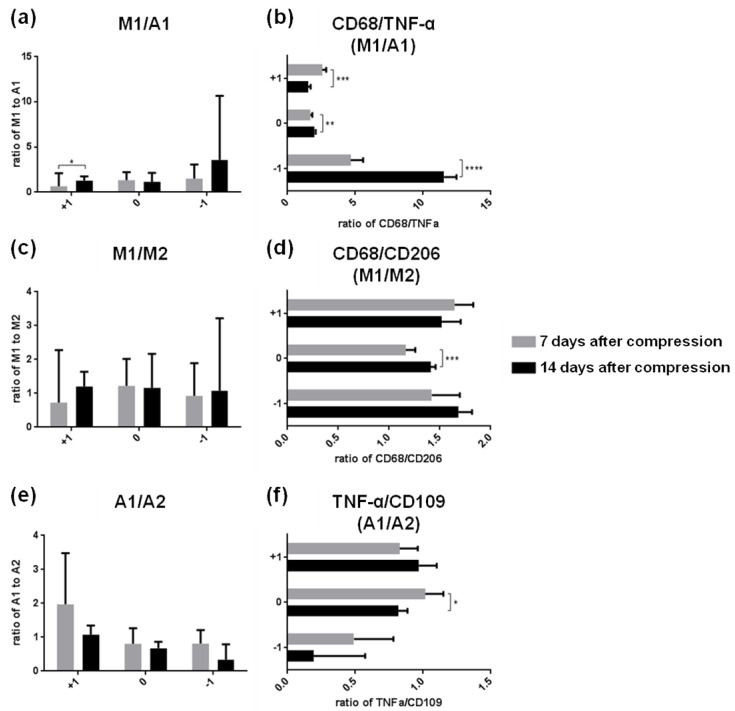
Microglia–astrocytes crosstalk at the injury site and in surrounding areas one and two weeks after SCI. (**a**,**b**) Graphs show ratios between overall neurotoxic M1 microglia markers and pro-inflammatory A1 astrocytes and their strongest representative markers (CD68 and TNF-α). (**c**,**d**) Ratios between neurotoxic *vs* neuroprotective phenotypes of microglia M1/M2 and their strongest representative markers (CD68 and CD206). (**e**,**f**) Ratios between neurotoxic *vs* neuroprotective phenotypes of astrocytes A1/A2 and their strongest representative markers (TNF-α/CD109). Error bars represent SD. Statistical significance was determined with parametric *t*-test (* *p* ˂ 0.05; ** *p* ˂ 0.01; *** *p* ˂ 0.001; **** *p* ˂ 0.0001). SCI—spinal cord injury; CD68—cluster of differentiation 68; TNF-α—tumor necrosis factor-α; CD206—mannose receptor and C-type lectin; CD109—Cluster of differentiation 109.

**Figure 8 cells-10-01943-f008:**
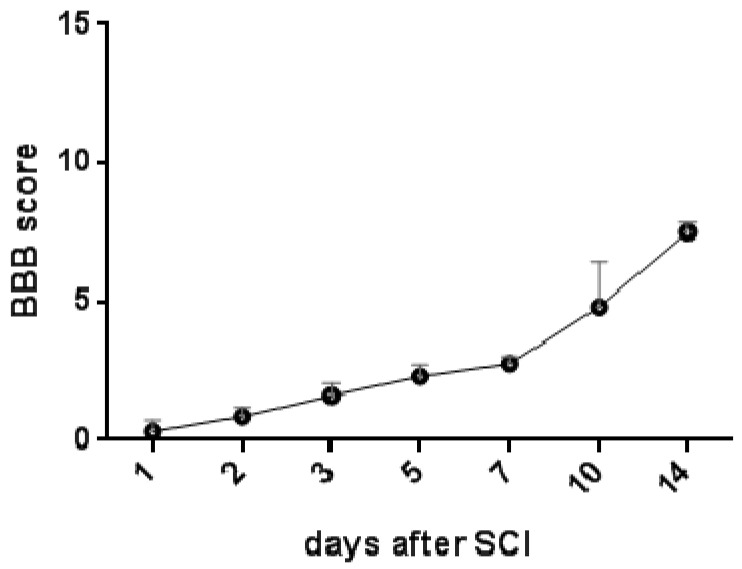
Graph showing locomotor activity of experimental animals after Th9 compression, evaluated with BBB score Scheme 21. (normal hindlimb movement), were recorded daily for the first three days and thereafter on day 5, 7, 10 and 14. Results are presented as mean ± SD (*n* = 5). BBB—Basso–Beattie–Bresnahan score.

**Figure 9 cells-10-01943-f009:**
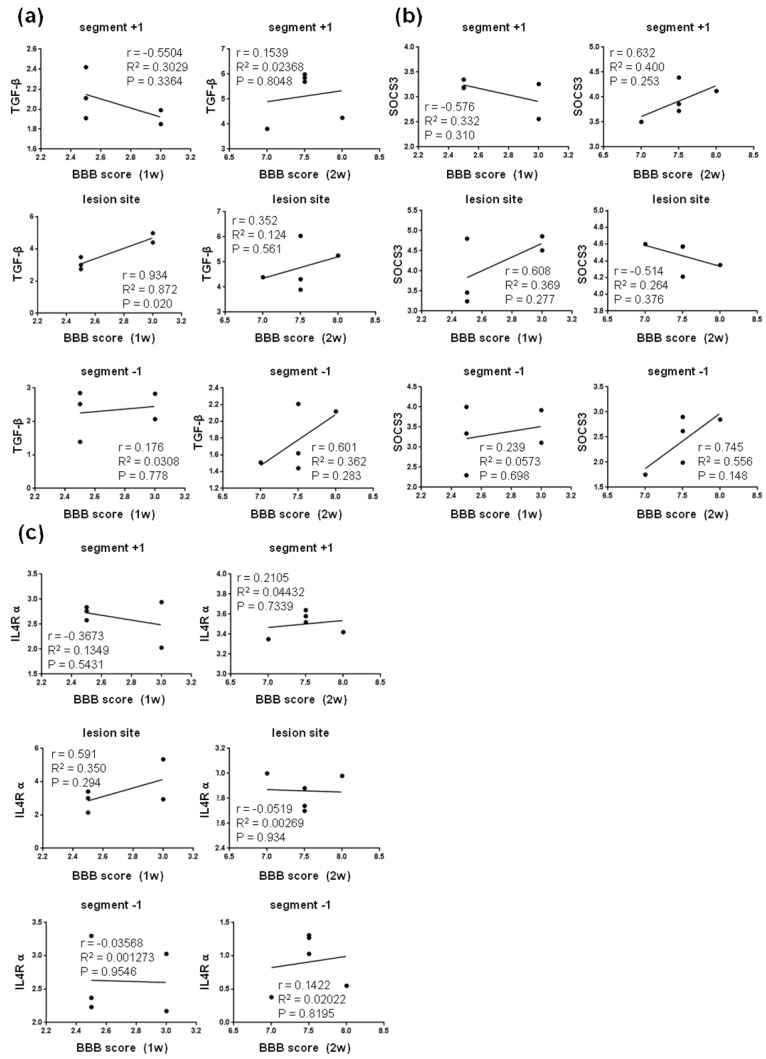
Linear regression analysis of correlation between BBB score and relative gene expression of markers characteristic for protective microglial M2c phenotypes—TGF-β (**a**), SOCS3 (**b**), and IL4Rα (**c**) at injury site and in adjacent cranial and/or caudal segments one and two weeks post-SCI. Scatterplots of individual values (*n* = 5) with regression line correlation coefficient (r), coefficients of determination (r2) calculated using regression analysis and *p*-value. SCI—spinal cord injury; TGF-β—transforming growth factor beta; SOCS3—suppressor of cytokine signaling 3; IL4R α—interleukin 4 receptor alpha.

**Figure 10 cells-10-01943-f010:**
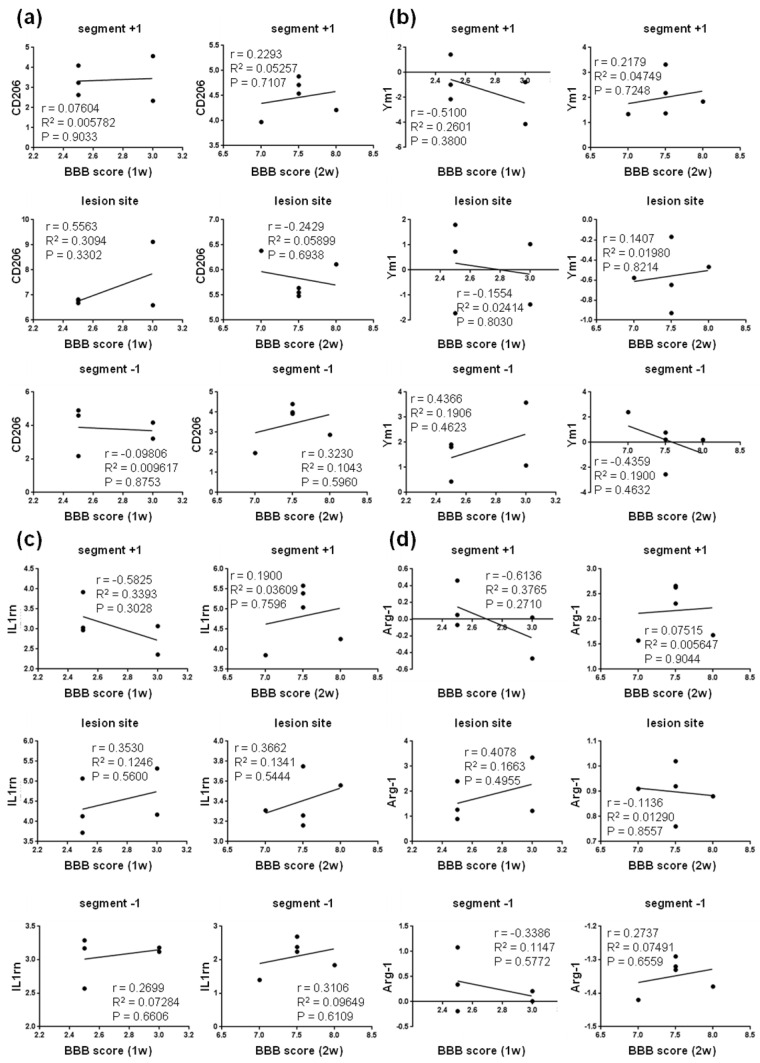
Linear regression analysis of correlation between BBB score and relative gene expression of markers characteristic for protective microglial M2a phenotypes—CD206 (**a**), Ym1 (**b**), IL1rn (**c**) and Arg-1 (**d**) at injury site and in adjacent cranial and/or caudal segments one and two weeks post-SCI. Scatterplots of individual values (*n* = 5) with regression line correlation coefficient (r), coefficients of determination (r2) calculated using regression analysis and *p*-value. CD206—mannose receptor and C-type lectin; Ym1—chitinase-like protein-1; IL1rn—interleukin 1 receptor antagonist; Arg-1—arginase-1.

**Figure 11 cells-10-01943-f011:**
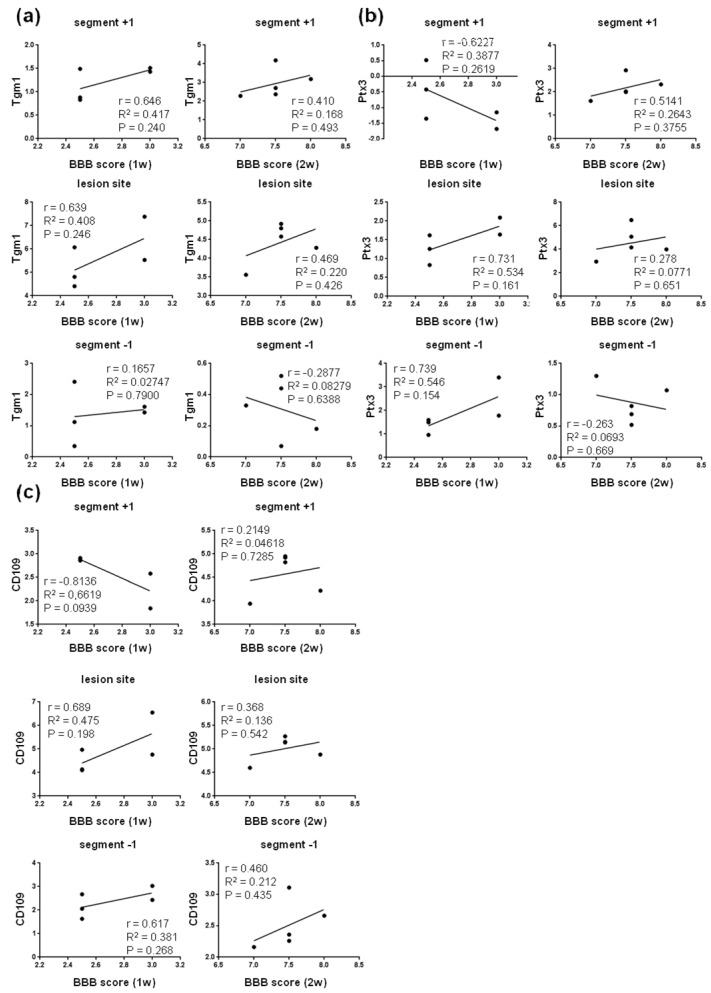
Correlation analysis of relative gene expression of neuroprotective A2 astrocyte markers Tgm1 (**a**), Ptx3 (**b**) and CD109 (**c**) in the cranio–caudal extent of the spinal cord versus BBB neurological scores at one and two weeks post-SCI. Scatterplots of individual values (*n* = 5) with regression line correlation coefficient (r), coefficients of determination (r2) calculated using regression analysis and *p*-value. SCI—spinal cord injury; Tgm1—Transglutaminase 1; Ptx3—Pentraxin 3; CD109—Cluster of differentiation 109.

**Table 1 cells-10-01943-t001:** Primers used in PCR study.

Target	Forward Primer (5’-3’)	Reverse Primer (5’-3’)
*18s RNA*	GACCATAAACGATGCCGACT	GTGAGGTTTCCCGTGTTGAG
*Arg-1*	CGGTGGCCTTTCTCCTGAA	GCAAGCCGATGTACACGATG
*C1q*	CCTCTTGTGTTTGGGGTCCC	CGTGTCAGCAGGGGACAGT
*C3*	GGAAGTGTTGTGAGGATGGCA	CTGATGAAGTGGTTGAAGACGG
*CD109*	GCTGCTTATGCGTTGCTCG	GGCCCTTTGATGGTCACTTG
*CD11b*	TGAAGAGCACCATCTGGGAC	AGATGGCGTACTTCACAGGC
*CD206*	AAGGTTCCGGTTTGTGGAG	TGCATTGCCCAGTAAGGAG
*CD68*	TCATGGGAATGCCACAGTTTC	GAGGGCCAACAGTGGAGAA
*Cx3Cr1*	TCCCGGAATTGGATCTAGAG	GCAGGACCTCGGGGTAATCA
*GFAP*	CAGCTTCGAGCCAAGGAG	TGTCCCTCTCCACCTCCA
*Iba1*	ATCCCAAGTACAGCAGTGATGAGGA	AAATAGCTTTCTTGGCTGGGGGAC
*IL1*β	TGACCCATGTGAGCTGAAAG	AGGGATTTTGTCGTTGCTTG
*IL1rn*	GGGAAAAGACCCTGCAAGA	GTGGATGCCCAAGAACACA
*IL4R*α	TCCGCACTTCTACGTGTGAG	AGACCACAGTTCCAGCCAGT
*IL-6*	CAGGAACGAAAGTCAACTCCA	ATCAGTCCCAAGAAGGCAACT
*iNOS*	GCTACGCCTTCAACACCAA	GCTTGTAACCACCAGCAGT
*Lcn2*	CTGTCTGTCTGCCGCTCCAT	AAGAGGGATCAGATGCTTGGTG
*Ptx3*	TGGTGGGTGGGAAGGAGAA	TGGCCATCTCCAGAGTGGTA
*S100B*	TTGCCCTCATTGATGTCTTCCA	TCTGCCTTGATTCTTACAGGTGAC
*Serpina3n*	TGCAAAACTGGACCCTCTGA	GCCTCAGGAGAAGCATCAACT
*SOCS3*	GGGACCAAGAACCTACGCAT	GGCTGCTCCTGAACCTCAAA
*TGF-*β	ATACGCCTGAGTGGCTGTC	GCCCTGTATTCCGTCTCCT
*Tgm1*	GCTCGAAGGTTCTGGGTTACA	TGGGAAAGCTGTGGACTGTC
*TNF*α	GCCCACGTCGTAGCAAAC	GCAGCCTTGTCCCTTGAA
*Ym1*	TGGAGGCTGGAAGTTTGGATC	CCACGAGACCCAGGGTATTG

## Data Availability

The datasets generated during and analyzed during the current study are available from the corresponding author on reasonable request.

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
