# Peer review of "Activation of Neuroprotective Microglia and Astrocytes at the Lesion Site and in the Adjacent Segments Is Crucial for Spontaneous Locomotor Recovery after Spinal Cord Injury"

_cells, 2021, doi:10.3390/cells10081943_

Round 1

Reviewer 1 Report

Review form

Paper to be reviewed: Microglial/astroglial interactions - an important therapeutic target in treatment of spinal cord injury

The paper investigates the prevalence of pro-inflammatory M1 microglia and A1 astrocytes versus anti-inflammatory M2 microglia and A2 astrocytes at the spinal cord level after a crush injury, at  the injury site, above (cranial) and below (caudal) it and at two time points (1 and 2 weeks post-injury). The results are nice and many correlations between the data were explored, but I found that the way in which the results are presented is somehow confusing and the main message is not very clearly sent through. More precisely:

  1. The title is more a review title. It should convey your main observation about which cell type is predominant at a certain lesion site and time point. From your data is not very clear how the putative interaction between microglia and astrocytes, which is not documented by other signaling molecules between them, could represent a therapeutic target. This is and extrapolation. Stick to the facts.
  2. Abstract is OK, although a summarizing sentence at the end might help. At the moment you end with a sentence about just another observation.
  3. Introduction and M&M section are clear, although table 1 is fuzzy – you might want to change it. Please explain why did you choose compression at Th9.
  4. Results

As indicated in the abstract I would have expected to have a clear picture at what is happening at the lesion site at the two time points, above it and below it. I think your paper might benefit from a heat map or a table where the reader should see at a glance what is happening at the lesion site, above it, below it at each time point, with microglia and astrocytes.

You should add control data as well, in all your bar graphs. You can’t talk about comparisons with control conditions if you don’t not show them.

Figure 1a. You measured the expression of Iba1, Cd11b and Cx3cr1. All are markers for MG/macrophages but it is interesting that they do not seem to be correlated at the same time points and locations, which is somehow puzzling. I would recommend to comment on it in the discussion section. What are the additional information you may get, for your story, from measuring the expression of all these markers?

Figure 2- why the decrease a GFAP and S100B at the core lesion, one week after the lesion, should suggest a rapid astrocyte death? Do you have a reference that this is happening?

Figure 8 – according to rows 369-371, neuroprotective MG are activated at all sites (lesion, cranial, caudal). Still, you do not show correlations for the chosen markers at all time points and locations, except for SOCS3. Please do so. Similarly for astrocytes (Figure 9).

  1. Discussion

Rows 410-41 our results show that the activation of unique neuroprotective processes between microglia and astrocytes takes place at least at the lesion site and in the adjacent cranial segment one week post-injury. You didn’t show any of these neuroprotective processes between MG and astrocytes; they might be there because you find some positive correlations, but you didn’t actually proven them. Rephrase.

You should separate your ideas in more paragraphs to make it easier to follow, for example paragraphs about CD11b, CD68…

Reviewer 2 Report

In the manuscript ‘Microglia/astroglial interactions- an important therapeutic target in the treatment of spinal cord injury’ the authors Kisucká et al., have described changes in the gene expression of different inflammatory and neurotrophic markers in the glial cells post- Spinal Cord Injury (SCI). They have shown, with supporting evidence, that activation of neuroprotective markers is required for spontaneous locomotor recovery. The authors have studied events after SCI between the first and second week, which is a crucial time window for pro- and anti-inflammatory signaling. The data presented shed light on the temporal and spatial progression of cellular changes that happen post-SCI. Knowledge from the study about the timeline of glial scar formation in experimental models is crucial for choosing the right treatment at a precise time after SCI. The study is well designed, and the data support the conclusions in the manuscript. Therefore this article can be a good addition to the literature.

Author Response

Thank you for your comment.

Round 2

Reviewer 1 Report

Thank you for your answers and the updated version of the paper. I will recommend publication with one amendment: some of the references about spinal cord compression at Th9 level should be introduced in the introduction section, to better define the purpose of the study.